# SPWOOD: SPARSE PARTIAL WEAKLY-SUPERVISED ORIENTED OBJECT DETECTION

**Wei Zhang[1], Xiang Liu[1], Ningjing Liu[1], Mingxin Liu[1], Wei Liao[2], Chunyan Xu[2], Xue Yang[1]***

[1]School of Automation and Intelligent Sensing, Shanghai Jiao Tong University
[2]Nanjing University of Science and Technology
Code: https://github.com/VisionXLab/SPWOOD

## ABSTRACT

A consistent trend throughout the research of oriented object detection (OOD) has been the pursuit of maintaining comparable performance with fewer and weaker annotations. This is particularly crucial in the remote sensing domain, where the dense object distribution and a wide variety of categories contribute to prohibitively high costs. Based on the supervision level, existing OOD algorithms can be broadly grouped into fully supervised, semi-supervised, and weakly supervised methods. Within the scope of this work, we further categorize them to include sparsely supervised and partially weakly-supervised methods. To address the challenges of large-scale labeling, we introduce the first Sparse Partial Weakly-Supervised Oriented Object Detection (SPWOOD) framework, designed to efficiently leverage only a few sparse weakly-labeled data and plenty of unlabeled data. Our framework incorporates three key innovations: (1) We design a Sparse-annotation-Orientation-and-Scale-aware Student (SOS-Student) model to separate unlabeled objects from the background in a sparsely-labeled setting, and learn orientation and scale information from orientation-agnostic or scale-agnostic weak annotations. (2) We construct a novel Multi-level Pseudo-label Filtering (MPF) strategy that leverages the distribution of model predictions, which is informed by the model's multi-layer predictions. (3) We propose a unique sparse partitioning approach, ensuring equal treatment for each category. Extensive experiments on the DOTA-v1.0/v1.5 and DIOR datasets show that SPWOOD framework achieves a significant performance gain over traditional OOD methods mentioned above, offering a highly cost-effective solution.

## 1 INTRODUCTION

In the field of oriented object detection (OOD) task, early research is often supervised by rotated bounding box (RBox) (Ding et al., 2019; Xie et al., 2021; Yang et al., 2019b; 2021), as shown in Figure 1(a). However, the dense distribution and diverse nature of objects in the remote sensing domain make it extremely difficult to obtain large-scale datasets with such detailed annotations.

To mitigate the reliance on fully annotated data, significant developments have been proposed, such as semi-supervised oriented object detection (SOOD) (Hua et al., 2023; Liu et al., 2021; 2022; Wang et al., 2025) and weakly supervised oriented object detection (WOOD) (Yang et al., 2023; Yu et al., 2025a; Luo et al., 2024) shown in Figure 1(b-c). Semi-supervised methods utilize pseudo-labeling strategies (Li et al., 2022a; Wang et al., 2023b) to learn angle and scale information from plenty of unlabeled data. Weakly supervised methods use training data with less detailed labels, such as horizontal bounding box (HBox) (Yu et al., 2023) or point (Yu et al., 2024; Ren et al., 2024; Zhang et al., 2025). More recently, two notable subfields have emerged that integrate these approaches to further the reduce annotation burdens: Partial weakly-supervised oriented object detection (PWOOD) (Liu et al., 2025a) integrate the unlabeled and weakly annotated datasets. Sparsely supervised oriented object detection (SAOD) (Suri et al., 2023; Rambhatla et al., 2022; Lu et al., 2024) leverage the labeled datasets containing annotations for only a fraction of the objects in an image. As illustrated in Figure 1(d-e), these methods alleviate the annotation dilemma further.

---

*Corresponding Author

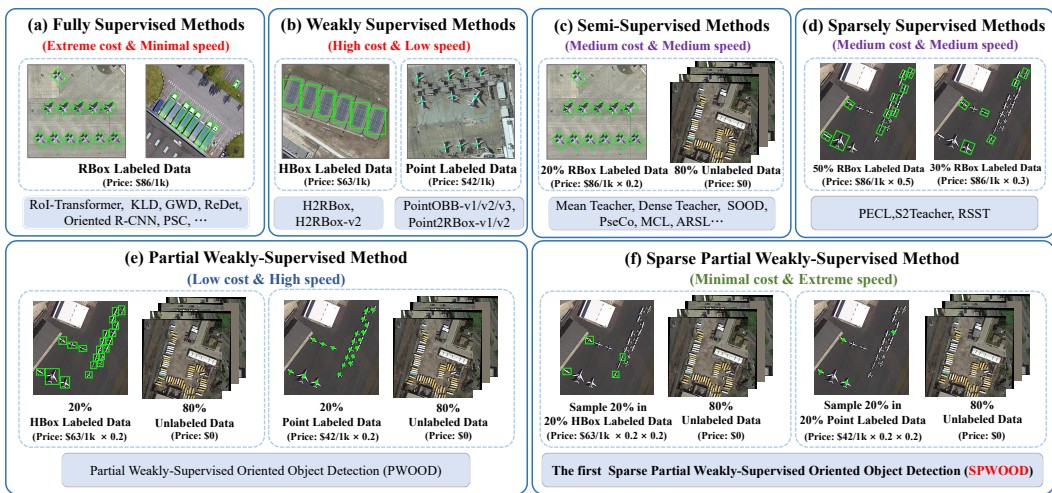

Figure 1: Current oriented object detection methods are predominantly classified into five categories. Compared to the aforementioned approaches, our proposed Sparse Partial Weakly-supervised Oriented Object Detection (SPWOOD) distinguished with minimal annotation requirements.

To further reduce annotation costs, we first propose a novel framework called Sparse Partial Weakly-supervised Oriented Object Detection (SPWOOD). This framework effectively leverages sparsely and weakly annotated data, along with unlabeled data. Inspired by the teacher-student paradigm (Tarvainen & Valpola, 2017), we leverage a small amount of sparsely and weakly annotated data for pre-training. Through this process, the student module learns the scale and angle information from weak annotations and acquires the ability to distinguish unlabeled object from background in the sparse annotation setting. As training progresses, the teacher module's capabilities are continuously enhanced through Exponential Moving Average (EMA) mechanism. Upon entering the unsupervised learning stage, the teacher module generates pseudo-labels for unlabeled data. These pseudo-labels serve as a strong supervisory signal to train the student module, allowing it to learn from both the limited sparsely and weakly annotated data and the abundant unlabeled data. Consequently, pseudo-labels' quality and the strategy used to filter them are crucial for model's performance.

Traditional semi-supervised methods for pseudo-label selection typically depend on static thresholds (Liu et al., 2021; Wang et al., 2025), which often leads to performance that is sensitive across different training processes (Chen et al., 2022a; Wang et al., 2023b; 2022; Zhong et al., 2020). In contrast, PWOOD (Liu et al., 2025a) introduced a more robust approach by leveraging Gaussian Mixture Model (GMM) to cluster the teacher model's predictions. While sparse-annotation methods, such as $S^2$Teacher (Lin et al., 2025) and RSST (Liao et al., 2025), employed sophisticated techniques like top-k high-confidence proposal selection or a class-aware label assignment mechanism that leverages the distribution of class features. To better utilize the teacher module's predictions in sparsely annotated setting, we designe a Multi-level Pseudo-labels Filtering (MPF) mechanism. Considering the inconsistencies between different layers, we dynamically adjust the filtering threshold for each layer's selection. This allows the model to adaptively generate more stable pseudo-labels that are better aligned with the teacher's performance. Our approach improves the model's ability to handle diverse and sparse scenarios, ultimately leading to more robust detection performance.

When creating sparse datasets, prior methods (Lu et al., 2024; Lin et al., 2025) follows what we term the Single Sparse Method, where annotations are processed on an image-by-image basis. A critical limitation of this technique is its inherent bias: when an image contains an annotation from a rare category, at least on annotation will be preserved simply. This disproportionately retains sparse categories and leads to a significant mismatch between the distribution of the processed data and the original dataset. To overcome this issue, we propose a novel approach called the Overall Sparse Method, treating all labeled annotations across the entire dataset as a single, unified group. This allows us to apply a consistent sampling ratio to each category, ensuring that every class is treated equally. As a result, our method effectively maintains the overall distribution of the original dataset. The contributions of this work are as follows:

- To our best knowledge, we introduce the first Sparse Partial Weakly-supervised Oriented Object Detection (SPWOOD) framework. This unified training pipeline is designed to robustly support various multi-format annotations (RBox, HBox, Point) or their combination as input under sparse partial annotation setting, thereby alleviating the significant burden of large-scale annotation.

- We construct a student model designed to acquire the crucial ability to perceive sparse annotation environments and learn object orientation and scale information within the sparse partial weakly-annotated scenarios. We term this model the SOS-Student model.

- The Multi-level Pseudo-labels Filtering (MPF) mechanism is designed to resolve the inconsistency between traditional selection criteria and the model's prediction confidence. By doing so, MPF significantly enhances the robustness of the pseudo-label selection process for unlabeled data.

- We propose a fundamentally distinct sparse annotation approach for creating sparse-partial datasets. Our SPWOOD framework has been rigorously trained and validated on the DOTA-v1.0/v1.5 and DIOR sparse-partial settings. SPWOOD achieves performance highly comparable to state-of-the-art Oriented Object Detection (OOD) algorithms.

## 2 RELATED WORK

### 2.1 SEMI-SUPERVISED ORIENTED OBJECT DETECTION

To effectively leverage abundant unlabeled data, the teacher-student framework has been widely adopted in semi-supervised object detection (Li et al., 2022b; Liu et al., 2023; Nie et al., 2023; Sun et al., 2021a). This method begins by training a student module on a limited set of fully labeled data. A teacher module then acquires the ability to generate pseudo-labels for unlabeled data by using an exponential moving average (EMA) of the student's weights. These pseudo-labels subsequently guide the student's training, creating a learning loop. For instance, MCL (Wang et al., 2025) introduced a novel approach by introducing Gaussian Center Assignment for labeled data and Scale-aware Label Assignment for unlabeled data. Besides, SOOD++ (Liang et al., 2024) treated remote sensing images as global layouts, explicitly establishing a many-to-many relationship between sets of pseudo-labels and predictions to enhance detection. However, despite their innovations, these methods still rely on a substantial number of fully annotated RBox for their initial training.

### 2.2 WEAKLY SUPERVISED ORIENTED OBJECT DETECTION

Weakly supervised object detection algorithms (Bilen et al., 2015; Iqbal et al., 2021; Yang et al., 2019a; Zhang et al., 2021; Zhu et al., 2023), as a significant breakthrough, offer a more efficient alternative to fully supervised methods by leveraging weak annotation, such as HBox or point annotations. For HBox-supervised methods (Li et al., 2022c; Sun et al., 2021b; Tian et al., 2021; Wang et al., 2024; Zhu et al., 2023), H2RBox (Yang et al., 2023) introduced a weakly supervised branch and a self-supervised branch to learn both scale and orientation information. Building on this, H2RBox-v2's primary innovation lay in its symmetry-based self-supervised learning (Yu et al., 2023), which directly derived crucial directional information from an object's inherent symmetry. For point-supervised methods, the main challenge is how to accurately learn object's scale and angle information. Recent studies have made significant progress in this area (Chen et al., 2021; 2022b; He et al., 2023; Ying et al., 2023). P2RBox (Cao et al., 2023) and PointSAM (Liu et al., 2025b) demonstrated remarkable performance by leveraging the zero-shot capabilities of the Segment Anything Model (Kirillov et al., 2023). Additionally, PointOBB (Luo et al., 2024) and its subsequent versions (Ren et al., 2024; Zhang et al., 2025) pushed the boundaries of point-supervised detection by using instance learning to solve the scale problem and class probability map to acquire angle information. More recently, Point2RBox (Yu et al., 2024) adopted a knowledge combination strategy by introducing synthetically generated targets, offering prior scale imformation. Furthermore, Point2RBoxv2 (Yu et al., 2025a) incorporated novel losses based on spatial layout constraints, which ensure that predictions align more accurately with real-world object. Besides, Wholly-WOOD (Yu et al., 2025b), a unified weakly supervised detector, accommodated multiple annotation formats including Point/HBox/RBox or their combination as inputs. More specifically, PWOOD (Liu et al., 2025a) merged the advantages of semi-supervised method and weakly supervised mthod to achieve enhanced performance, all while substantially lowering annotation requirements.

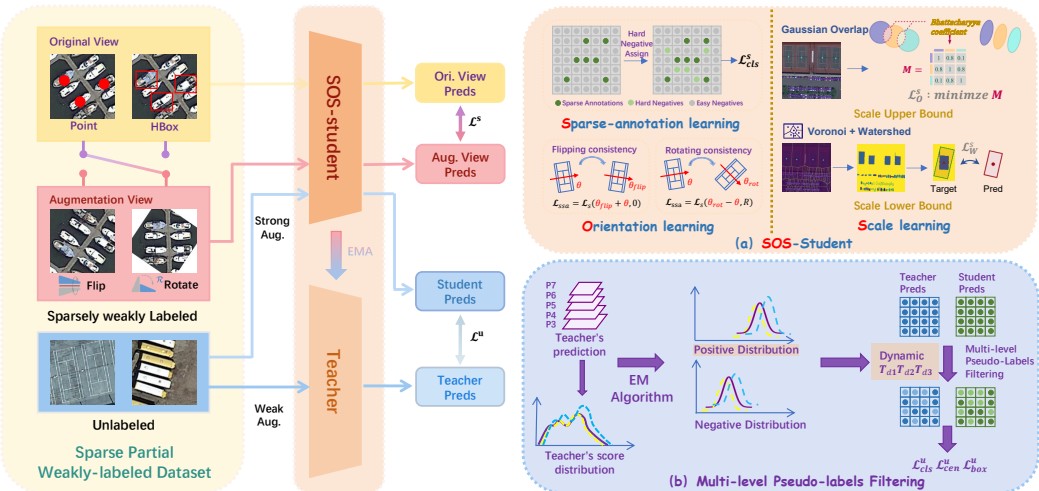

Figure 2: The illustration of the Sparse Partial Weakly-supervised Oriented Object Detection (SPWOOD). The Sparse-annotation-Orientation-and-Scale-aware Student (SOS-Student) identifies hard negatives and learn the scale and angle information from sparse weak annotation data. The Multi-level Pseudo-labels Filtering (MPF) mechanism acquires ability from student through EMA algorithm and selects high-quality pseudo-labels for student module's training.

## 2.3 SPARSELY SUPERVISED ORIENTED OBJECT DETECTION

In SAOD research filed, only a fraction of objects in an image are labeled, while the rest remains unlabeled. Lack of complete annotations presents a major challenge, as the detector confuse unlabeled objects with the background, since both share the same "background" label. In the domain of 3D object detection, CoIn (Xia et al., 2023) introduced a contrastive learning module to enhance feature discrimination and a feature-level pseudo-label mining framework to guide the training. HINTED (Xia et al., 2024) proposed a self-boosting teacher, leveraging instance-level information and updates pseudo-labels for labeled scenes to enhance learning effectiveness. When addressing similar challenges within the remote sensing domain, Co-mining (Wang et al., 2021) utilized a Siamese network to enhance multi-view learning, with two branches predicting pseudo-labels for each other via a co-generation module. Region-based approaches (Rambhatla et al., 2022) treated SAOD as a semi-supervised problem at the region level, focusing on identifying unlabeled regions that likely contain foreground objects. Calibrated Teacher (Wang et al., 2023a) introduced an online calibration mechanism to fit the true precision during training, improving pseudo-labels quality. PECL (Lu et al., 2024) offered a reinforcement learning-based selection strategy specifically tailored for pesudo-labels filtering. More recently, S$^2$Teacher (Lin et al., 2025) proposed a clusterbased pseudo-label generation module to avoid erroneous guidance. RSST (Liao et al., 2025) designed a class-aware pseudo-labeling mechanism for both labeled and unlabeled data by integrating priors from large language model. In contrast to these methods, our approach innovatively integrates various supervised methods under minimal supervision cost.

## 3 METHOD

### 3.1 OVERVIEW

Given the setting consisting of abundant unlabeled data and a small amount of sparsely-weakly annotated data, we adopt a classic pseudo-labeling semi-supervised object detection (SSOD) framework. As shown in Figure 2, SPWOOD framework operates with two core branches: one for supervised learning and another for unsupervised learning. The supervised branch integrates sparse-annotation learning, scale learning, and orientation learning, which together form the SOS-Student module. The unsupervised branch leverages a Multi-level Pseudo-labels Filtering (MPF) algorithm to generate reliable supervisory signals used for student module's subsequent training.

The training process unfolds in two distinct stages: (1) Burn-in Stage: The student module begins by training on the a few sparse weak annotations from both original and augmented views. Concur-

rently, the learned weights of the SOS-Student module are mirrored onto the teacher module. (2) Self-training Stage: In this stage, a large volume of strongly-weakly-augmented unlabeled data is fed into the teacher module to generate pseudo-labels. The student module is then further optimized using these pseudo-labels. The teacher module is updated from the student's module using an Exponential Moving Average (EMA) (Tarvainen & Valpola, 2017) approach, which ensures the teacher's parameters are more stable and reliable, leading to higher-quality pseudo-labels.

## 3.2 SOS-STUDENT

### 3.2.1 SPARSE ANNOTATION LEARNING

In the context of sparsely annotated data, a major challenge is effectively distinguishing unlabeled objects from background. Both are treated as negatives during training, leading to potential misguidance where unannotated objects are incorrectly penalized as background. Inspired by Focal Loss, designed to tackle the class imbalance problem by down-weighting the loss of easy-to-classify examples, we introduce a mechanism to differentiate between different types of negative samples by modulating their loss contribution (Liao et al., 2025). This strategy aims to balance the influence of false supervision and retain beneficial background cues. The loss function is formulated as follows:

$$\mathcal{L}_{cls}^s = \begin{cases} -\alpha_t(1-p_t)^\gamma \log(p_t) & \text{for positive objects} \\ -(1-\alpha_t)p_t^\gamma \log(1-p_t) & \text{for negative objects with confidence } p_t \le thr \\ -(1-\alpha_t)p_t^\gamma \log(1-p_t)\omega & \text{for negative objects with confidence } p_t > thr, \end{cases} \quad (1)$$

Our student classification loss, $L_{cls}^s$, aims to partition the prediction space into three distinct groups for effective learning under sparse supervision:Labeled Objects: Predictions with high confidence ($p_t > thr$) matching the ground truth (GT) labeled category.Background: Predictions with low confidence ($p_t \le thr$) matching the GT background.Unlabeled Objects (Hard Negatives): Predictions with high confidence ($p_t > thr$) that are erroneously matched to the GT background.The base loss formulation incorporates standard components: $p_t$ represents the confidence score from the student module; $\alpha_t$ serves as a balancing factor between positive and negative samples; and $\gamma$ is the focusing parameter, designed to reduce the contribution of well-classified examples (inheriting the robustness of Focal Loss).For the critical third group (Unlabeled Objects), we introduce the adaptive factor $\omega$. This factor strategically down-weights these potentially misleading false negatives.This novel strategy effectively inherits the robustness of Focal Loss while simultaneously providing a targeted solution for sparse annotation scenarios. By mitigating the detrimental effects of misleading false negatives, it leads to more accurate and robust model learning.

### 3.2.2 ORIENTATION LEARNING

To address the lack of orientation information in weakly-annotated data, like HBox annotations, we introduce a symmetry-aware learning approach (Yu et al., 2023) that comprehensively explores the properties of object symmetry. As depicted in Figure 2, each input image undergoes a random augmentation (either flipping or rotating) to create an augmented view. This view is then fed into the student module, which generates a pair of predicted angles. Since the input views have a clear relationship, the output angles are expected to follow the same relationship.

To ensure the SOS-Student effectively learns orientation information from the weakly annotated horizontal bounding boxes, we formulate an angle loss, $\mathcal{L}_{Ang}^s$

$$\mathcal{L}_{Ang}^s = \begin{cases} L_{Ang}^s(\theta_{flp} + \theta, 0) & \text{if augmentation is a flip} \\ L_{Ang}^s(\theta_{rot} - \theta, \mathcal{R}) & \text{if augmentation is a rotation by } \mathcal{R}. \end{cases} \quad (2)$$

The angle loss calculation depends on the specific image augmentation method model used, which can be either a vertical flip or a random rotation by an angle $\mathcal{R}$. The loss function $L_{Ang}^s$ is a Smooth-L1 loss. Here, $\theta_{flp}$, $\theta_{rot}$, and $\theta$ represent the predicted angles of the flip-augmented image, the rotated-augmented image, and the original image, respectively.

Table 1: Comparison with state-of-the-art methods for the OOD task at different sparse-partial ratios. $\triangle$10% and $\blacksquare$10% denote the partial-ratio and sparse-ratio. RSST* represents our SAOD baseline and R:H:P means the annotations count ratio between RBox, HBox and Point.

| Algorithm Types | Methods | Annotations | DOTA-v1.0 Dataset-Sparse-Partial | | | | | |
| --- | --- | --- | --- | --- | --- | --- | --- | --- |
| | | | 10%$\triangle$ | | 20% | | 30% | |
| | | | 10%$\blacksquare$ | 20% | 10% | 20% | 10% | 20% |
| Weakly Supervised | H2RBox-v2 | HBox | 30.6 | 30.8 | 38.5 | 42.7 | 43.9 | 49.2 |
| | Point2RBox-v2 | Point | 9.4 | 9.8 | 14.2 | 23.1 | 15.1 | 27.0 |
| Semi-Supervised | MCL | RBox | 31.7 | 39.0 | 37.6 | 44.5 | 43.5 | 47.8 |
| | PWOOD | RBox | 38.0 | 46.2 | 46.2 | 51.9 | 48.7 | 55.2 |
| Partial Weakly-Supervised | PWOOD | HBox | 33.9 | 43.8 | 42.4 | 47.6 | 44.8 | 50.7 |
| | PWOOD | Point | 17.0 | 24.7 | 22.4 | 28.6 | 23.1 | 33.8 |
| Sparsely Supervised | S$^2$Teacher | RBox | 36.8 | 44.0 | 45.3 | 50.2 | 52.3 | 55.5 |
| | RSST | RBox | 43.4 | 47.2 | 42.5 | 52.3 | 53.0 | 56.6 |
| | RSST* | RBox | 42.4 | 45.5 | 41.7 | 51.0 | 53.2 | 56.5 |
| **Sparse Partial Weakly-Supervised** | **SPWOOD (ours)** | RBox | **48.5** | **54.0** | **54.9** | **57.8** | **54.9** | **60.3** |
| | | HBox | **45.5** | **51.9** | **52.2** | **54.0** | **53.1** | **56.5** |
| | | Point | **27.3** | **36.8** | **33.3** | **38.7** | **35.8** | **41.8** |
| | | R:H:P=1:1:1 | 42.4 | 48.2 | 46.1 | 53.0 | 50.8 | 54.8 |

### 3.2.3 SCALE LEARNING

Given that weaker annotations, such as point annotations, they lack crucial scale and orientation information about the objects. To address this, we adopt spatial layout learning (Yu et al., 2025a) to learn the objects' scale information by determining both the upper and lower bounds.

For the upper bound of the object's scale, we minimize the distance between different predicted oriented bounding boxes. We first model these boxes as two-dimensional Gaussian distributions and then use the Bhattacharyya distance (Yang et al., 2022) to calculate the distance between them. This allows us to derive the Gaussian overlap loss, $\mathcal{L}_O^s$, as follows:

$$\mathcal{L}_O^s = \frac{1}{N}\sum_{i \neq j} B(\mathcal{N}_i, \mathcal{N}_j), \tag{3}$$

where $N$ donates the number of predicted rotated bounding boxes, $\mathcal{N}_i$ and $\mathcal{N}_j$ represent different Gaussian distribution, and $B$ is the Bhattacharyya distance between them.

To determine the lower bound of the object's scale, we introduce the Voronoi Watershed Loss. A Voronoi diagram (Aurenhammer, 1991) separates the entire image into individual regions, ensuring each region contains only one point annotation. These regions are then fed into a watershed algorithm (Vincent & Soille, 1991) to obtain a pixel-level classification based on pixel similarity. By rotating the output of the watershed algorithm to align with the direction of predicted rotated bounding boxes based on a single point, we can obtain the regression target of object's width and height. The Voronoi watershed loss, $\mathcal{L}_W$, is then designed to regress the width and height of the objects:

$$\mathcal{L}_W^s = L_{GWD}\left(\begin{bmatrix} w/2 & 0 \\ 0 & h/2 \end{bmatrix}^2, \begin{bmatrix} w_t/2 & 0 \\ 0 & h_t/2 \end{bmatrix}^2\right), \tag{4}$$

where $L_{GWD}$ is Gaussian Wasserstein Distance Loss (Yang et al., 2022).

Finally, we incorporate class loss $\mathcal{L}_{cls}^s$, centerness loss $\mathcal{L}_{cen}^s$, and box loss $\mathcal{L}_{box}^s$. The total supervised loss, $\mathcal{L}^s$, for the SOS-Student model is given by:

$$\mathcal{L}^s = w_{cls}\mathcal{L}_{cls}^s + w_{cen}\mathcal{L}_{cen}^s + w_{box}\mathcal{L}_{box}^s + w_{Ang}\mathcal{L}_{Ang}^s + w_O\mathcal{L}_O^s + w_W\mathcal{L}_W^s, \tag{5}$$

where $\mathcal{L}_{cls}^s$, $\mathcal{L}_{cen}^s$, and $\mathcal{L}_{box}^s$ represent the sparse annotation aware loss (as described in 3.2.1), the cross entropy loss and the IoU loss respectively. The weights $w_{cls}, w_{cen}, w_{box}$ are set to 1, while $(w_{Ang}, w_O, w_W)$ are set to (0.2, 10, 5) by default.

Table 2: Comparison of mAP performance across different methods for the OOD task on DOTA-v1.5 test set under different sparse-partial ratios (RBox supervised).

| Algorithm Types | Methods | 10-10 | 10-20 | 20-10 | 20-20 | 30-10 | 30-20 |
|---|---|---|---|---|---|---|---|
| Semi-Supervised | MCL | 25.9 | 33.9 | 33.2 | 40.2 | 38.2 | 42.9 |
| | PWOOD | 33.9 | 38.1 | 40.9 | 45.9 | 45.1 | 49.2 |
| Sparsely Supervised | S$^2$Teacher | 33.8 | 37.2 | 41.1 | 45.7 | 45.7 | 46.2 |
| | RSST | 37.1 | 40.8 | 42.3 | 46.9 | 45.3 | 48.0 |
| | RSST$^*$ | 36.2 | 35.8 | 42.3 | 46.0 | 45.7 | 49.8 |
| **Sparse Partial Weakly-Supervised** | **SPWOOD (ours)** | **43.2** | **47.9** | **49.0** | **52.1** | **51.3** | **53.1** |

## 3.3 MULTI-LEVEL PSEUDO-LABELS FILTERING

The selection of pseudo-labels directly influences the performance of subsequent training. In SAOD setting, RSST (Liao et al., 2025) leverages classes'diversity and LLM'assistance to select pseudo-labels, combining a fixed number of predictions per category and overall predictions. However, ignoring the confidence variance throughout the training process of model makes the detector's performance sensitive. Furthermore, Feature Pyramid Networks (FPN) (Lin et al., 2017) is specialized for detecting objects at corresponding scales and the same object exhibits different confidence scores at different levels (*i.e.* P3, P4, P5, P6, P7), making modeling the entire prediction directly unreliable as Class-Agnostic PseudoLabel Filtering strategy (CPF) (Liu et al., 2025a).

To better utilize the information from the model's prediction and discover reliable pseudo-labels, we focus on the distribution of the teacher's predictions by introducing Multi-level Pseudo-labels Filtering. Based on a Gaussian Mixture Model (GMM) (Wang et al., 2023b; Zhao et al., 2019), we model the prediction confidence of each layer in the teacher module with the following equation:

$$\mathcal{P}^i(c^i) = w_p^i \mathcal{N}_p^i(\mu_p^i, (\sigma_p^i)^2) + w_n^i \mathcal{N}_n^i(\mu_n^i, (\sigma_n^i)^2), \tag{6}$$

where $\mathcal{P}^i(c^i)$ means the modeling of $i-$th level, such as P3, P4, P5, P6 and P7. $\mathcal{N}(\mu, \sigma^2)$ denotes gaussian distribution, while $w_p$, $\mu_p$, $\sigma_p$ and $w_n$, $\mu_n$, $\sigma_n$ represent the weight, mean and variance of positive and negative distributions, respectively. The GMM is initialized by setting $\mu_p$ and $\mu_n$ to the maximum and minimum of the predicted scores, respectively. The variances ($\sigma_p$ and $\sigma_n$) and weights ($w_p$ and $w_n$) are all initially set to 1 and 0.5, respectively. We then use Expectation-Maximization (EM) algorithm to solve for the posterior probability, $\mathcal{P}$, with the following equation:

$$\tau^i = \arg\max_{c^i} \mathcal{P}^i(c^i, \mu_p^i, (\sigma_p^i)^2), \tag{7}$$

where $\tau^i$ is then used to select pseudo-labels at the corresponding scale level. The selected pseudo-labels from each layer are subsequently used to guide the student module's training.

## 3.4 OVERALL LOSS

Our proposed SPWOOD framework contains two branches: one for the supervised loss $\mathcal{L}^s$ and another for the unsupervised loss $\mathcal{L}^u$. The combination of these two losses constitutes the overall loss. The former one is detailed in 3.2.3. The latter one is defined as below:

$$\mathcal{L}^u = \mathcal{L}_{cls}^u(\mathcal{T}^c, \mathcal{S}^c) + \mathcal{L}_{cen}^u(\mathcal{T}^{cen}, \mathcal{S}^{cen}) + \mathcal{L}_{box}^u(\mathcal{T}^{logit}, \mathcal{S}^{logit}), \tag{8}$$

where $\mathcal{T}$ and $\mathcal{S}$ represent the predictions of the teacher and student modules, respectively. These prediction include the confidence score ($c$), centerness ($cen$) and the margin from the point to the boundaries of the predicted boxes. The loss function $\mathcal{L}_{cls}^u$ and $\mathcal{L}_{cen}^u$ are binary cross-entropy losses, while $\mathcal{L}_{box}^u$ is Smooth-L1 loss. The overall loss of SPWOOD framework is defined as:

$$\mathcal{L} = \mathcal{L}^s + \mathcal{L}^u. \tag{9}$$

Through these two complementary branches, the student learns from sparsely-weakly annotated data, enhancing the teacher's pseudo-labels filtering ability. In turn, the high-quality pseudo-labels selected by teacher further improve the student's learning, forming a positive feedback learning loop.

Table 3: Comparison of mAP performance across different methods for the OOD task on DIOR test set under different sparse-partial ratios (RBox supervised).

| Algorithm Types | Methods | 10-20 | 20-20 | 30-20 |
|---|---|---|---|---|
| Semi-Supervised | MCL | 30.8 | 33.1 | 35.8 |
| | PWOOD | 31.0 | 36.8 | 39.1 |
| Sparsely Supervised | S²Teacher | 36.1 | 42.6 | 45.1 |
| | RSST | 40.7 | 44.8 | 46.1 |
| | RSST* | 38.8 | 43.2 | 45.7 |
| **Sparse Partial Weakly-Supervised** | **SPWOOD (ours)** | **44.1** | **45.7** | **46.3** |

Table 4: Detection accuracy under different combinations of weak annotations at the sparse-partial ratio 20-20.

| RBox:HBox:Point | mAP |
|---|---|
| 1:1:1 | 53.0 |
| 1:1:0 | 56.3 |
| 1:0:1 | 52.3 |
| 0:1:1 | 47.6 |
| 0:1:4 | 41.2 |

Table 5: mAP results of Sparse Annotation Learning with different weights under sparse-partia 20-20 ratio.

| Weight | mAP |
|---|---|
| 0.4 | 57.8 |
| 0.3 | 57.5 |
| **0.2** | **60.6** |
| 0.1 | 58.0 |

Table 6: Comparison of mAP performance across Sparse Annotation Learning (SAL) module in our framework in DOTA1.0 test set under sparse-partia 20-20 ratio.

| Setting | mAP |
|---|---|
| with SAL | 47.6 |
| without SAL | 42.0 |

## 4 EXPERIMENT

### 4.1 DATASET AND SETUP

To evaluate our proposed SPWOOD models, we conducted experiments on DOTA-v1.0/-v1.5 (Xia et al., 2018) and DIOR (Li et al., 2020). DOTA-v1.0 comprises 2,806 aerial images, consisting of 1,411 training images, 458 validation images, and 937 test images. The training and validation sets contain 188,282 instances across 15 categories. DOTA-v1.5 uses the same images as DOTA-v1.0 but features more annotations, with 403,318 instances across 16 categories, including a higher number of smaller objects. The DIOR dataset comprises 23,463 images across 20 distinct object categories. The dataset is partitioned into 11,725 images for training and 11,738 images for testing.

Taking the DOTA-V1.0 dataset as a representative example, to create sparse-partial weak supervision dataset, we select 10%, 20%, and 30% of the images as initial labeled data from the DOTA-v1.0 train-val set, while the remaining images are treated as unlabeled. From the labeled subset, we generate datasets using two distinct sparse methods. The first is the Single Sparse Method (Lu et al., 2024; Lin et al., 2025), which applies a specific sparse ratio (*i.e.* 10%, 20%, 30%) to each category within each image, keeping at least one annotation for any category present in the image. In contrast, our proposed Overall Sparse Method treats all labeled annotations as a unified group and samples annotations for each category at the desired ratio (*i.e.* 10%, 20%, 30%). For a fair comparison with prior studies, we conduct main experiments on the datasets generated by the Single Sparse Method, unless otherwise noted. Then we create the weakly-annotated data for training by simply omitting the orientation and scale information from the annotations. In our naming convention, we combine the partial-ratio and sparse-ratio (*i.e.* 30-10), where 30% indicates the partial-ratio and 10% indicates the sparse-ratio. Besides, all reported detection results of models are obtained by testing on test set.

Our proposed SPWOOD model is implemented using the MMRotate (Zhou et al., 2022) frameworks. We employ an FCOS detector (Tian et al., 2019) with a ResNet50 backbone (He et al., 2016) and a FPN (Lin et al., 2017) neck. The AdamW optimizer (Loshchilov & Hutter, 2017) is used for optimization. The entire training schedule consists of 180,000 iterations, which includes an initial burn-in stage of 12,800 iterations to stabilize model convergence in the early stage.

### 4.2 MAIN RESULT

DOTA-v1.0: We employ a simplified version of RSST, comprising both supervised and unsupervised branches, as our SAOD baseline, which we term RSST*. As summarized in Table 1, we present the detection results of different methods on DOTA-v1.0 dataset and SPWOOD exhibits a substantial

Table 7: Detection accuracy at sparse-partial ratio 20-50 (RBox supervised).

| Algorithm Types | Methods | 20-50 |
|---|---|---|
| Semi-Supervised | MCL | 53.2 |
| | PWOOD | 59.8 |
| Sparsely Supervised | $S^2$Teacher | 56.5 |
| | RSST | 56.1 |
| | RSST* | 55.1 |
| **Sparse Partial Weakly-Supervised** | **SPWOOD (ours)** | **63.0** |

Table 8: The performance comparisons of our proposed PWOOD framework based on different Pseudo-labels Filtering Methods at different parse-partial ratios under DOTA-v1.0 Dataset-Sparse-Partial setting.

| Module | Methods | 10-10 | 10-20 | 20-10 | 20-20 |
|---|---|---|---|---|---|
| SPWOOD | w/ CPF | 44.4 | 53.0 | 51.9 | 57.1 |
| | **w/ MPF** | **49.5** | **54.0** | **54.9** | **57.8** |

Table 9: Performance of the PWOOD Framework with Overall and Single Sparse Methods.

| Algorithm paradigms | Methods | 10-10 | 10-20 | 20-10 | 20-20 | 30-10 | 30-20 |
|---|---|---|---|---|---|---|---|
| SPWOOD | Overall Sparse Method | 41.7 | 49.1 | 47.9 | 57.2 | 52.3 | 57.5 |
| | Single Sparse Method | 49.5 | 54.0 | 54.9 | 57.8 | 54.9 | 60.3 |

and consistent improvement over other methods across all sparse-partial ratios. Notably, SPWOOD achieves superior performance using the less-informative HBox annotations, even outperforming the RSST* model that utilizes corresponding proportions of RBox annotations with gains of 3.1% (10-10), 6.4% (10-20), 10.5% (20-10) and 3.0% (20-20). It means that SPWOOD delivers excellent performance at a lower cost. The effectiveness of SPWOOD under weak supervision is further highlighted by comparisons with WOOD methods. Compared to H2RBox-v2, SPWOOD achieves large margins of improvement, yielding gains of 14.9% (10-10), 21.1% (10-20), 13.7% (20-10), 11.3% (20-20), 9.2% (30-10), and 7.3% (30-20) under the corresponding HBox annotations. Similarly, under point annotations, SPWOOD improves mAP by 17.9%, 27.0%, 19.1%, 15.6%, 20.7%, and 14.8%, respectively, compared to Point2RBox-v2. These results strongly underscore SPWOOD's excellent capability in the sparse-partial, weakly-supervised oriented object detection task.

Concurrently, SPWOOD's ability to support multiple annotation formats within a unified framework offers a highly effective paradigm for reducing the data acquisition burden. As detailed in Table 1 (*i.e.* R:H:P=1:1:1), we conducted experiments to evaluate the framework's performance under diverse labeling scenarios, using a combination of Point, HBox, and RBox annotations as input. The detection performance significantly surpasses WOOD and SOOD methods. Crucially, SPWOOD achieves highly competitive performance compared to the current State-of-the-Art SAOD methods across different sparse-partial ratios. In Table 4, we present comprehensive results from experiments conducted under different mixed weak annotation settings. This analysis empirically confirms the robust capability of our framework to accommodate and leverage different types of weak supervision concurrently. Furthermore, as detailed in Table 7, we present the results at the ratio 20-50, highlighting SPWOOD's effectiveness in utilizing limited annotation resources.

More Results: As shown in Table 2, to provide additional validation for our proposed framework's efficacy, we performed a thorough comparative analysis under RBox annotations on DOTA-v1.5 dataset. Our proposed SPWOOD outperforms RSST* with gains of 7% (10-10), 12.1% (10-20), 6.7% (20-10), 6.1% (20-20), 5.6% (30-10) and 3.3% (30-20), respectively. Compared to the gain on DOTA-v1.0 (6.1%, 8.5%, 13.2%, 6.8%, 1.7%, and 3.8%, respectively), SPWOOD's performance on DOTA-v1.5 highlights its robustness and adaptability in complex scenes with smaller objects. To validate the capability of our model across different datasets, we conducted additional experiments on the DIOR dataset. As demonstrated in Table 3, our model consistently outperforms all competing methods across all the ratios.

## 4.3 ABLATION STUDIES

The selection of pseudo-labels directly impacts the model's subsequent training, making the filtering strategy crucial. As detailed in Table 8, we conducted a rigorous systematic evaluation of our proposed Multi-level Pseudo-label Filtering (MPF) algorithm against the Class-Agnostic Pseudo-label Filtering (CPF) approach (Liu et al., 2025a). Our MPF method consistently outperforms CPF across all tested sparse-partial ratios on the DOTA-v1.0 dataset. Specifically, our MPF method achieves notable mAP improvements of 5.1% (10-10), 1.0% (10-20), 3.0% (20-10), and 0.7% (20-20), respectively. The most significant gains are observed at lower annotation sparsity levels (e.g., 10-10),

Table 10: Annotation Statistics and Performance Analysis for Overall and Single Sparse Methods at sparse-partial ratio 10-10. 20-10 and 30-10. ▲ and ▼ mean annotations numbers under Single and Overall Sparse Method. ■ presents relative difference among corresponding numbers.

| Category | PL | BD | BR | GTF | SV | LV | SH | TC | BC | ST | SBF | RA | HA | SP | HC |
|---|---|---|---|---|---|---|---|---|---|---|---|---|---|---|---|
| Annotation at 30-10 | 762▲ | 88 | 211 | 242 | 2083 | 1342 | 2344 | 238 | 81 | 376 | 137 | 156 | 514 | 225 | 40 |
|  | 785▼ | 37 | 114 | 39 | 2163 | 1319 | 2316 | 189 | 44 | 389 | 34 | 40 | 567 | 124 | 25 |
| Relative Difference | -2.9%■ | +137.8 | +85.1 | +520.5 | -3.7 | +1.7 | +1.2 | +25.9 | +84.1 | -3.3 | +302.9 | +290.0 | -9.3 | +81.5 | +60.0 |
| Annotation at 20-10 | 551▲ | 57 | 117 | 151 | 1256 | 863 | 1868 | 180 | 59 | 306 | 90 | 99 | 352 | 157 | 40 |
|  | 585▼ | 19 | 68 | 24 | 1270 | 873 | 1824 | 121 | 35 | 275 | 22 | 27 | 366 | 111 | 30 |
| Relative Difference | -5.8%■ | +200 | +72.1 | +529.2 | -1.1 | -1.1 | +2.4 | +48.8 | +68.6 | +11.3 | +309.1 | +266.7 | -3.8 | +41.4 | +33.3 |
| Annotation at 10-10 | 383▲ | 37 | 77 | 79 | 767 | 479 | 846 | 100 | 25 | 224 | 48 | 46 | 134 | 110 | 32 |
|  | 369▼ | 14 | 29 | 13 | 779 | 491 | 870 | 78 | 20 | 232 | 12 | 11 | 145 | 95 | 24 |
| Relative Difference | +3.8%■ | +164.3 | +165.5 | +507.7 | -1.5 | -2.4 | -2.8 | +28.2 | +25.0 | -3.4 | +300 | +318.2 | -7.6 | +15.8 | +33.3 |
| AP at 10-10 | 71.6▲ | 46.8 | 17.3 | 53.6 | 55.0 | 52.3 | 73.2 | 86.1 | 50.3 | 62.5 | 27.6 | 48.1 | 19.8 | 53.9 | 23.0 |
|  | 73.4▼ | 23.6 | 17.6 | 25.6 | 55.9 | 56.4 | 72.6 | 88.8 | 37.1 | 67.5 | 0.40 | 24.8 | 18.5 | 40.8 | 18.0 |
| Relative Difference | -2.5%■ | +98.6 | -1.7 | +109.4 | -1.6 | -7.2 | +0.8 | -3.1 | +35.5 | -7.5 | +583.0 | +94.2 | +6.9 | +32.1 | +27.6 |

demonstrating MPF's superior capability in extremely sparse settings. This superior performance stems from MPF's ability to effectively capture intricate relationships within the model's multi-level predictions, thereby achieving superior detection accuracy even with limited supervision.

Sparse Processing Method Analysis: To rigorously evaluate the impact of data generation on model performance, we conducted ablation study focusing on datasets created by two sparse methods. As shown in Table 9, the SPWOOD model demonstrates consistently superior performance on the dataset derived from Single Sparse Method compared to Overall Sparse Method, with gains of 7.8%, 4.9%, 7.0%, 0.6%, 2.6%, and 2.8% across the corresponding ratios. As detailed in Table 10, the Single Sparse Method tends to retain more annotations for categories with initially scarce labels, such as baseball diamond, ground track field, soccer-ball field, and roundabout. We further analyzed the direct relationship between annotation count and detection accuracy (Table 10). For instance, considering the baseball-diamond category under ratio 10-10, the Single Sparse Method yielded a 164.3% relative increase in annotation count compared to the Overall Sparse Method. This substantial data advantage directly translated into a near 100% (98.6%) relative difference on AP performance. These findings decisively reveal that relatively higher annotation count leads to superior detection accuracy, which is particularly pronounced for categories with few initial annotations.

More results: We conducted an ablation study on the parameter $w$ within our Sparse Annotation Learning module, as shown in Table 5. In our framework, the primary function of $w$ is to suppress the influence of hard negatives, ensuring robust training despite sparse annotations.Furthermore, we performed a ablation study about Sparse Annotation Learning module in the SOS-Student component, with results presented in Table 6. The substantial performance degradation observed after removing this module clearly underscores its critical contribution in the sparse setting. These results underscore the Sparse Annotation Learning module's efficacy in semi-sparse setting.

## 5 CONCLUSION

In this work, we propose a Sparse Partial Weakly-Supervised Oriented Object Detection (SPWOOD) framework, which leverages the strengths of mainstream oriented object detection methods to significantly reduce the need for annotation in remote sensing. Given the sparse weak annotation data, we introduce three core components—sparse annotation learning, orientation learning, and scale learning—to form SOS-Student model. To effectively utilize abundant unlabeled data, we employ a Multi-level Pseudo-labels Filtering mechanism to select reliable supervised signals. Extensive experiments on benchmark datasets demonstrate that SPWOOD outperforms existing OOD methods, all with a minimal annotation cost. Furthermore, we introduce an novel sparse processing method, guaranteeing that the sparse dataset maintains the same distribution as the original data. One current limitation of our approach is that it exclusively utilizes a single visual modality. Incorporating additional data modalities (e.g., spectral or textual information) is posited as an interesting future research direction, particularly for further improving performance within sparse annotation settings.

## ACKNOWLEDGMENT

This work was supported by the National Natural Science Foundation of China (62506229), Natural Science Foundation of Shanghai under 25ZR1402268, and Shanghai QiYuan Innovation Foundation.

## ETHICS STATEMENT

This work is conducted in accordance with all ethical standards. We have appropriately cited all compared and referenced methods. The datasets used are publicly available and have been appropriately cited. We have ensured that our data collection and processing methods, as well as the release of our code, do not violate privacy rights or intellectual property laws.

## REPRODUCIBILITY STATEMENT

To ensure the reproducibility of our work, we have provided comprehensive details of our experimental setup, methodology, and results. This includes a clear description of the algorithms, model architectures, hyperparameters, and training procedures used. We will make our source code, including all necessary scripts and configurations, publicly available upon publication. Besides, we will provide detailed instructions on how to prepare sparse partial weak supervision dataset.

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

Table 11: Comparison of computational costs across different methods for the OOD task on DIOR test set under different sparse-partial ratios (RBox supervised).

| Algorithm Types | Methods | memory usage | running time |
|---|---|---|---|
| Semi-Supervised | MCL | 5598MB | 16hours |
| Partial Weakly-Supervised | PWOOD | 9021MB | 23hours |
| Sparsely Supervised | RSST | 13672MB | 34hours |
| Sparse Partial Weakly-Supervised | SPWOOD (ours) | 22785MB | 40hours |

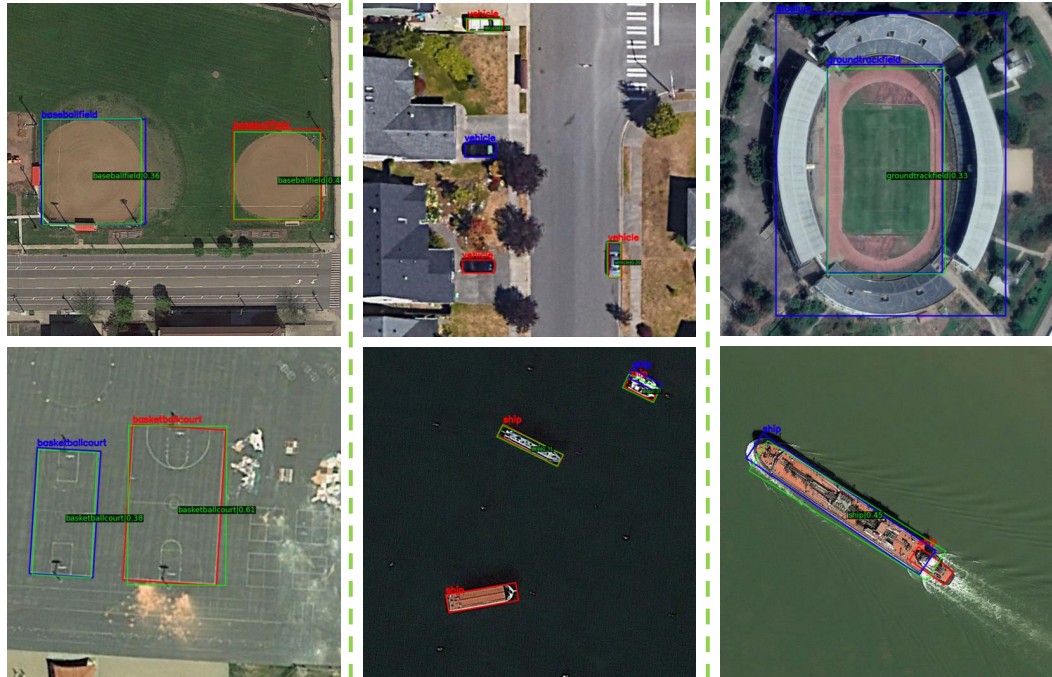

Figure 3: Qualitative results showing the qualities of the detection performance.

# A    APPENDIX

## A.1    THE USE OF LARGE LANGUAGE MODELS (LLMS)

We affirm that this paper is prepared and written entirely by us. We did not use any large language models (LLMs) to generate the abstract, content, or any part of the text. The ideas, analysis, and conclusions presented are the sole product of the authors' original thought and research. We did, however, utilize standard tools like grammar checkers for minor stylistic improvements.

## A.2    COMPUTATIONAL COST COMPARISON

As shown in Table 11, the operational cost (time and computational resource consumption) of our spwood method is observed to be higher compared to other models. This increase is primarily attributed to two factors: The necessary overhead associated with processing multiple weak annotations within the model, like combination of HBox and point. The substantial computational load required for effectively handling the highly sparse annotation settings, where extensive calculations are performed to compensate for the lack of dense supervision.

## A.3 FAILURE ANALYSIS

As depicted in Figure 3, we use different colors to illustrate the different bounding boxes under sparse conditions: the blue boxes represent the sparse annotations used for training; the red boxes indicate objects that were omitted from the labels due to the sparse setting; and the green boxes represent the final objects detected by our SPWOOD framework. The figure highlights the following scenarios and challenges:

Column 1 (Effectiveness in Sparse Setting): This column demonstrates our model's effectiveness in sparse scenarios, showing its ability to learn class features from partially labeled data and successfully detect other unlabeled instances of the same category in image.

Column 2 (Intra-class Variance Challenge): This column highlights cases where the model fails to detect certain objects. This failure is typically due to significant intra-class variance (differences within the same category) present in the sparsely annotated data.

Column 3 (Complex Spatial Arrangement Challenge): This section illustrates the performance degradation observed when dealing with complex spatial arrangements, such such as large objects containing smaller ones or closely packed, overlapping instances.

