# OpenReview forum: "SPWOOD: Sparse Partial Weakly-Supervised Oriented Object Detection"
_ICLR.cc/2026/Conference — ICLR 2026 Poster_

### Official Review · Reviewer_Xk5r · 2025-10-20

**Soundness:** 4
**Presentation:** 4
**Contribution:** 4
**Rating:** 8
**Confidence:** 5

**Summary:**

The paper proposes the SPWOOD (Sparse Partial Weakly-Supervised Oriented Object Detection) framework, which is the first to support multi kinds of annotations, such as RBox, HBox and point under a sparse partial weakly-labeled setting, forming a unified training pipeline for oriented object detection (OOD) task.
The SPWOOD framework is built upon the classic Teacher-Student paradigm, aiming to efficiently utilize a minimal amount of sparsely and weakly labeled data alongside a large volume of unlabeled data. The core innovations include three aspects:

1.	SOS-Student (Sparse-annotation-Orientation-and-Scale-aware Student) Model: This model is specifically designed for sparse annotation scenarios, effectively distinguishing between unlabeled objects and background. It also employs self-supervised learning to extract crucial orientation and scale information from weak annotations that initially lack these details.

2.	Multi-level Pseudo-label Filtering (MPF) Mechanism: To overcome the limitations of conventional pseudo-label filtering strategies (e.g., static thresholds or single-distribution modeling), MPF utilizes the Gaussian Mixture Model (GMM) to model the confidence distribution of predictions at every layer of the Feature Pyramid Network (FPN). This allows for dynamic adjustment of filtering thresholds, leading to the generation of higher quality and more stable pseudo-labels.

3.	Overall Sparse Method: An innovative sparse dataset creation technique. It treats all annotations across the entire dataset as a whole, applying a consistent sampling ratio to each category. This ensures that all classes are treated equally and effectively maintains the original dataset's class distribution.

The paper conducts extensive experiments on the DOTA-v1.0 and v1.5 datasets under a sparse-partial annotation setting. The results demonstrate that the SPWOOD framework achieves significant performance improvements compared to traditional semi-supervised and weakly supervised methods and reaches highly competitive performance with current state-of-the-art OOD algorithms.

**Strengths:**

1.	A Unified Framework: SPWOOD is the first OOD framework to support multi kinds of annotations, such as RBox, HBox and point under a sparse partial weakly-labeled setting, significantly pushing the boundaries of OOD research. It provides a theoretical and practical foundation for remote sensing applications where labeled data is extremely scarce.

2.	Strong Robustness to Sparsity: The SOS-Student model effectively addresses the core challenge in sparse annotation—the misclassification of unlabeled objects as background (Hard Negatives)—by introducing an adaptive loss modulation mechanism. This improves the model's learning accuracy in sparsely annotated scenarios.

3.	High-Quality Pseudo-label Generation: The MPF mechanism accounts for the prediction score discrepancies across different FPN levels by independently modeling the distribution for each layer. This allows for dynamic and targeted filtering of pseudo-labels, which significantly enhances the quality and stability of the guidance signal during the semi-supervised learning phase.

4.	Fair Data Sampling Mechanism: The proposed Overall Sparse Method ensures a fair sampling process by applying a consistent sampling ratio to all classes across the entire dataset. This avoids the disproportionate preservation of rare categories often seen in traditional Single-Image Sparse Methods, ensuring the class distribution of the sparse data remains consistent with the original dataset.

5.	Superior Performance: The experimental results demonstrate that SPWOOD achieves highly competitive performance on the DOTA datasets, validating the effectiveness of the proposed methodology.

**Weaknesses:**

1. Missing Bounding Box Parameter Definition: In the context of the bounding box loss (Formula 4), the crucial dimensional parameters w and h are not explicitly defined. It is unclear whether these variables refer to the predicted box, anchor box.

2. The sparse processing and the semi processing, it is unclear whether the data is first made sparse and then split, or vice versa. This opacity in the data pre-processing pipeline severely hampers the reader's ability to accurately replicate the structure of the final SPWOOD dataset.

3. Authors should incorporate a dedicated discussion about the inherent constraints of the current research and promising avenues for future work.

**Questions:**

1. Regarding Parameter Definitions in Formulas (4): In the bounding box loss (Formula 4), please explicitly define the symbols w and h?. Are these variables referring to the dimensions of the predicted box, the anchor box?

2. Could the authors provide a flowchart or a detailed textual description clarifying the precise order of execution between the sparse processing and the semi processing during the SPWOOD dataset generation?

3. Given that the point annotation scenario provides the most minimal supervision, have the authors considered integrating the knowledge or capabilities of large vision foundation models (e.g., SAM) in future work?

4. To enhance the paper's academic completeness and facilitate future research, the authors are requested to incorporate a dedicated discussion about inherent constraints of the SPWOOD framework and promising avenues for future development.

---

> ### Author Response · Authors · 2025-11-21
>
> > Weaknesses 1: Missing Bounding Box Parameter Definition: In the context of the bounding box loss (Formula 4), the crucial dimensional parameters w and h are not explicitly defined. It is unclear whether these variables refer to the predicted box, anchor box.
>
> Thank you for this suggestion. In formula 4, w and h present the predicted box, while w_t and h_t present the anchor box.
>
> > Weaknesses 2: The sparse processing and the semi processing, it is unclear whether the data is first made sparse and then split, or vice versa. This opacity in the data pre-processing pipeline severely hampers the reader's ability to accurately replicate the structure of the final SPWOOD dataset.
>
> Thank you for this insightful observation. When generating semi-sparse dataset, we first split the whole dataset into labeled sets and unlabeled set. Then we apply sparse method to labeled set.
>
> > Weaknesses 3: Authors should incorporate a dedicated discussion about the inherent constraints of the current research and promising avenues for future work.
>
> Thank you for this practical suggestion. Our proposed framework trackles the issues in remote sensing domain with only visual modality dataset. It can be further developed to contain other modality dataset.

---

> ### Author Response · Authors · 2025-11-21
>
> > Questions 1: Regarding Parameter Definitions in Formulas (4): In the bounding box loss (Formula 4), please explicitly define the symbols w and h?. Are these variables referring to the dimensions of the predicted box, the anchor box?
>
> Thank you for this suggestion. As mentioned in our response to Weakness 1, we have added a detailed description in the revised manuscript.
>
> > Questions 2: Could the authors provide a flowchart or a detailed textual description clarifying the precise order of execution between the sparse processing and the semi processing during the SPWOOD dataset generation?
>
> Thank you for this question. As mentioned in our response to Weakness 2, we have added a detailed description in the revised manuscript.
>
> > Questions 3: Given that the point annotation scenario provides the most minimal supervision, have the authors considered integrating the knowledge or capabilities of large vision foundation models (e.g., SAM) in future work?
>
> Thank you for this insightful observation. Considering SAM’s fully supervised pretraining, it conflicts with our weak-supervision setup.
>
> > Questions 4: To enhance the paper's academic completeness and facilitate future research, the authors are requested to incorporate a dedicated discussion about inherent constraints of the SPWOOD framework and promising avenues for future development.
>
> Thank you for this practical suggestion. As mentioned in our response to Weakness 3, we conduct analysis about constraints and future work in the revised manuscript.

---

### Official Review · Reviewer_JpkM · 2025-10-27

**Soundness:** 3
**Presentation:** 3
**Contribution:** 2
**Rating:** 4
**Confidence:** 3

**Summary:**

This paper proposes SPWOOD, a Sparse Partial Weakly-supervised Oriented Object Detection framework that reduces annotation costs by jointly leveraging sparse, weakly annotated, and unlabeled data. The method introduces three main components: a Sparse-annotation-Orientation-and-Scale-aware Student (SOS-Student) to learn orientation and scale from weak annotations, a Multi-level Pseudo-label Filtering (MPF) strategy to dynamically select reliable pseudo-labels, and an Overall Sparse Method ensuring balanced category sampling. Overall, the paper is clearly written and easy to follow. However, the related work discussion and experimental scope could be more comprehensive, and the novelty relative to closely related sparse or partial weak supervision literature needs clearer differentiation.

**Strengths:**

1.The paper investigates a new and practical Sparse Partial Weakly-supervised Oriented Object Detection problem that reduces annotation effort in remote sensing.

2.The proposed SOS-Student effectively integrates orientation and scale learning under sparse and weak supervision.

3.The writing and structure are clear, logical, and easy to follow, with well-presented figures and tables.

4.The framework’s generality across multiple annotation types (RBox, HBox, Point) increases its potential practical significance.

**Weaknesses:**

1.The related work lacks discussion of similar sparse partial supervision paradigms explored in other domains, such as 3D detection (e.g., HINTED), which share comparable annotation reduction strategies and technical challenges.

2.The experimental validation is confined to DOTA datasets, limiting the evidence of generality; evaluation on other datasets or domains would strengthen the claims.

3.The baseline selection is relatively weak, with no comparison to stronger or more recent sparse partial supervision approaches in general object detection.

4.The paper lacks qualitative failure analysis, which would help identify common error patterns and highlight the limitations of SPWOOD under difficult or ambiguous scenes.

5.The claim of being the “first” to propose this setting seems overstated, as the method mainly adapts existing ideas from related supervision paradigms; broader validation, stronger baselines, and deeper technical discussion would make the contribution more convincing.

**Questions:**

1.How does SPWOOD perform on other datasets beyond DOTA to verify its generalization capability?

2.How do existing sparse partial supervision methods perform or adapt in oriented object detection, and how does SPWOOD differ from them technically?

3.Would including visual qualitative examples or failure cases help demonstrate how SPWOOD handles sparse or ambiguous annotations?

---

> ### Author Response · Authors · 2025-11-21
>
> > Weaknesses 1: The related work lacks discussion of similar sparse partial supervision paradigms explored in other domains, such as 3D detection (e.g., HINTED), which share comparable annotation reduction strategies and technical challenges.
>
> Thank you for this valuable suggestion. To enhance the context of our literature review, we have incorporated a detailed discussion of related sparse supervision methods from other domains into the Related Work section of the revised manuscript.
>
> > Weaknesses 2: The experimental validation is confined to DOTA datasets, limiting the evidence of generality; evaluation on other datasets or domains would strengthen the claims.
>
> Thank you for raising this important concern regarding the generality of our findings. To strengthen our claims and demonstrate the broader applicability of our method, we have incorporated additional experimental validation on the DIOR dataset.
>
> |Algorithm Types| Methods| 10-20| 20-20| 30-20|
> |--------------|----|----|----|----|
> |Semi-Supervised| MCL |30.8| 33.1| 35.8|
> |Semi-Supervised|PWOOD |31.0| 36.8| 39.1|
> |Sparsely Supervised|S2Teacher| 36.1 |42.6| 45.1|
> |Sparsely Supervised|RSST |40.7 |44.8| 46.1|
> |Sparsely Supervised|RSST∗ |38.8| 43.2| 45.7|
> |**Sparse Partial Weakly-Supervised**| **SPWOOD (ours)**| **44.1**| **45.7**|**46.3**|
>
>  and result is shown in table 3 on page 8 in the revised manuscript.
>
> > Weaknesses 3: The baseline selection is relatively weak, with no comparison to stronger or more recent sparse partial supervision approaches in general object detection.
>
> Thank you for your feedback regarding the baseline selection. we must clarify that our proposed SPWOOD is the first sparse partial weakly-supervised oriented object detection framework, designed to tackle weak annotation in the semi-sparse setting within the remote sensing domain.
>
> Consequently, the baseline we selected, RSST, represents the most relevant and state-of-the-art work currently available in the related sparse-supervised domain. By positioning SPWOOD against this recent and strong sparse supervision method, we are able to effectively highlight the unique advantages and performance gains delivered by our framework under the challenging semi-sparse setting.

---

> ### Author Response · Authors · 2025-11-21
>
> > Weaknesses 4: The paper lacks qualitative failure analysis, which would help identify common error patterns and highlight the limitations of SPWOOD under difficult or ambiguous scenes.
>
> Thank you for this valuable suggestion. We agree that a qualitative failure analysis is crucial for comprehensively understanding the limitations and error modes of our proposed method. In the revised manuscript, we have incorporated a qualitative failure analysis section in Appendix A.3.
>
> > Weaknesses 5: The claim of being the “first” to propose this setting seems overstated, as the method mainly adapts existing ideas from related supervision paradigms; broader validation, stronger baselines, and deeper technical discussion would make the contribution more convincing.
>
> Following an extensive and thorough survey of literature in sparse partial weakly-supervised oriented object detection field, we must clarify that our work represents a pioneering effort.
>
> To the best of our knowledge, we are the first to propose a unified algorithm specifically designed for the sparse-semi-supervised setting that robustly supports the combinations of multiple heterogeneous annotation formats.
>
> Furthermore, to ensure a comprehensive and rigorous experimental analysis, we have benchmarked our method against the related up-to-date supervised paradigms available in the literature.

---

> ### Author Response · Authors · 2025-11-21
>
> > Questions 1: How does SPWOOD perform on other datasets beyond DOTA to verify its generalization capability?
>
> As mentioned in our response to Weakness 2, We add additional experiment on DIOR dataset to address your concern and the result is shown in table 3 in the revised manuscript..
>
> > Questions 2: How do existing sparse partial supervision methods perform or adapt in oriented object detection, and how does SPWOOD differ from them technically?
>
> Existing methods primarily focus on reducing the volume of RBox annotations required:
> - Sparsely Supervised Methods: These approaches typically train on an entire dataset where only a fixed ratio of the objects are labeled with full RBox (Oriented Bounding Box) annotations. Their primary goal is to learn from limited full RBox labels.
> - Semi-Supervised Methods: These methods utilize a small amount of RBox-labeled data alongside a large pool of unlabeled data. They rely on pseudo-labeling or consistency regularization to exploit the unlabeled data.
>
> To some extent, both paradigms successfully mitigate the annotation burden. However, their fundamental limitation is that they only support and rely on RBox annotations as the source of ground truth and pseudo-labels, making the required labeling process still time-consuming.
>
> As mentioned in our response to Weakness 3, SPWOOD is the first sparse partial weakly-supervised oriented object detection framework, tackling the multi-format weak supervision challenge under semi-sparse setting.
>
> We would be grateful if you could share any relevant work or related literature that you think would be beneficial to our study. We are committed to thoroughly reviewing and learning from any material you provide.
>
> We look forward to your suggestions.
>
> > Questions 3: Would including visual qualitative examples or failure cases help demonstrate how SPWOOD handles sparse or ambiguous annotations?
>
> As previously mentioned in our response to Weakness 4, we have incorporated a dedicated qualitative analysis section in Appendix A.3 in the revised manuscript.

---

> > ### Comment · Reviewer_JpkM · 2025-11-28
> > **SPWOOD is the first sparse partial weakly-supervised oriented object detection framework**
> >
> > Thank you for your detailed responses, and I apologize for the late reply. At this stage I can no longer update my rating, but I would like to acknowledge several clarifications you provided.
> > First, I agree that the sparse partial weakly-supervised oriented object detection setting is indeed proposed for the first time in the remote sensing domain, and the problem formulation is meaningful given the annotation burden in oriented detection. The design of SPWOOD, especially the unified handling of heterogeneous weak annotations under a semi-sparse setting, appears technically sound and addresses a practical gap that has not been explored in prior remote sensing literature.
> > Your added experiments on DIOR, the expanded related work discussion covering sparse supervision paradigms in other domains, and the inclusion of qualitative failure analysis all help strengthen the submission. While some concerns remain regarding baseline breadth and generality, the overall contribution shows a certain degree of novelty and practical value.
> >
> > Therefore, although my original score cannot be changed at this point, my final stance is that I am inclined to support acceptance. The rating remains:
> > Rating: 6 marginally above the acceptance threshold.

---

> > > ### Author Response · Authors · 2025-11-28
> > >
> > > Dear Reviewer JpkM,
> > >
> > > Thank you very much for your thoughtful feedback and for maintaining your positive assessment of our work. We greatly appreciate your support.
> > >
> > > Best regards,
> > >
> > > Authors

---

### Official Review · Reviewer_2UVa · 2025-10-28

**Soundness:** 3
**Presentation:** 2
**Contribution:** 2
**Rating:** 4
**Confidence:** 4

**Summary:**

This paper introduces SPWOOD, a framework to tackle the novel and challenging task of Sparse Partial Weakly-Supervised Oriented Object Detection. This problem setting is highly relevant, especially for domains like remote sensing where annotation costs are prohibitive.

**Strengths:**

The primary contributions are: (1) defining this new, complex problem setting; (2) proposing a unified teacher-student framework that leverages sparse, weak (HBox/Point), and unlabeled data simultaneously; (3) introducing several components, including a Sparse-annotation-Orientation-and-Scale-aware Student (SOS-Student) and a Multi-level Pseudo-labels Filtering (MPF) mechanism. The experiments conducted on DOTA datasets demonstrate that the proposed framework achieves state-of-the-art performance, significantly outperforming methods from related sub-fields.

**Weaknesses:**

Despite the impressive results and the important problem formulation, I have several major concerns regarding the methodology's novelty and the paper's internal consistency.
1. Questionable Novelty of the Core Supervised Model (SOS-Student)
The paper presents the SOS-Student as a key innovation, but its core components for learning orientation and scale appear to be directly adopted from prior work.
•	Orientation Learning (Sec 3.2.2): This is explicitly stated to be a symmetry-aware approach from Yu et al. (2023).
•	Scale Learning (Sec 3.2.3): This adopts the spatial layout learning method from Yu et al. (2025a), using Gaussian overlap and Voronoi watershed losses.
While combining these techniques in a new framework is an engineering contribution, the paper presents them as foundational parts of its proposed model without clearly delineating its own novel algorithmic contribution. The main new component, Sparse Annotation Learning (Sec 3.2.1), is itself insufficiently described.
2. Lack of Crucial Implementation Details and Unverified Assumptions
The paper omits critical details for key methods, making them unreproducible and their effectiveness difficult to assess.
•	Sparse Annotation Learning: The proposed modification to Focal Loss hinges on an "adaptive factor $w$" to down-weight potential unlabeled objects. The paper provides no information on how $w$ is defined or computed. Is it a fixed hyperparameter, a function of the prediction confidence $p_t$, or something else? This is a crucial detail that is entirely missing.
•	Multi-level Pseudo-labels Filtering (MPF): This method is based on the strong assumption that the teacher's prediction scores at each FPN level can be effectively modeled by a Gaussian Mixture Model (GMM). The paper provides no empirical validation (e.g., visualizations of score distributions) to support this claim. If the distributions are heavily skewed or non-Gaussian, the GMM-based filtering may not be robust.
3. Significant Contradiction in the Experimental Design and Claimed Contributions
This is my most serious concern. The authors propose a new "Overall Sparse Method" for dataset creation as a contribution to overcome the bias of the existing "Single Sparse Method" (Lines 100-104, 375-377).
However, the paper's own ablation study in Table 5 clearly shows that their proposed "Overall Sparse Method" is consistently and significantly inferior to the "Single Sparse Method" across all settings. More problematically, the authors then abandon their own proposed method and use the superior-performing "Single Sparse Method" for all main comparison experiments (Table 1, 2), citing "a fair comparison with prior studies" (Line 394).
This creates a major logical flaw: a method introduced as a contribution is empirically proven to be worse and is subsequently discarded in the main experiments. A contribution that fails its own evaluation cannot be claimed as a strength of the paper and undermines the work's scientific rigor.

**Questions:**

None

---

> ### Author Response · Authors · 2025-11-21
>
> > Weaknesses 1: Questionable Novelty of the Core Supervised Model (SOS-Student) The paper presents the SOS-Student as a key innovation, but its core components for learning orientation and scale appear to be directly adopted from prior work. • Orientation Learning (Sec 3.2.2): This is explicitly stated to be a symmetry-aware approach from Yu et al. (2023). • Scale Learning (Sec 3.2.3): This adopts the spatial layout learning method from Yu et al. (2025a), using Gaussian overlap and Voronoi watershed losses. While combining these techniques in a new framework is an engineering contribution, the paper presents them as foundational parts of its proposed model without clearly delineating its own novel algorithmic contribution. The main new component, Sparse Annotation Learning (Sec 3.2.1), is itself insufficiently described.
>
> As you say, we acknowledge that our SOS-Student module integrates existing weak supervision algorithms. However, we must clarify that our contribution extends far beyond a simple combination of these algorithms.
>
> The fundamental challenge we address is the sparse supervision setting. We found that non-trivial fusion of existing weak supervision techniques yields poor results when faced with data scarcity.
>
> Therefore, our primary technical contribution is the specially-designed Sparse-Oriented Learning Module within the SOS-Student framework. This module introduces novel mechanisms necessary to robustly utilize multi-format weak supervision under extreme sparsity.
>
> We will provide a more thorough and detailed technical description of this Sparse-Oriented Learning Module in the revised manuscript to clearly demonstrate how it enables the entire framework to function effectively.

---

> ### Author Response · Authors · 2025-11-21
>
> > Weaknesses 2: Lack of Crucial Implementation Details and Unverified Assumptions The paper omits critical details for key methods, making them unreproducible and their effectiveness difficult to assess. • Sparse Annotation Learning: The proposed modification to Focal Loss hinges on an "adaptive factor $w$" to down-weight potential unlabeled objects. The paper provides no information on how $w$ is defined or computed. Is it a fixed hyperparameter, a function of the prediction confidence $p_t$, or something else? This is a crucial detail that is entirely missing. • Multi-level Pseudo-labels Filtering (MPF): This method is based on the strong assumption that the teacher's prediction scores at each FPN level can be effectively modeled by a Gaussian Mixture Model (GMM). The paper provides no empirical validation (e.g., visualizations of score distributions) to support this claim. If the distributions are heavily skewed or non-Gaussian, the GMM-based filtering may not be robust.
>
> Thank you for raising this important concern regarding the adaptive factor. We have fully addressed this point by conducting a dedicated ablation study focusing on the optimal definition and impact of this factor in our methodology.
>
> |Weight| mAP|
> |-------|-------|
> |0.4| 57.8|
> |0.3| 57.5|
> |**0.2**| **60.6**|
> |0.1| 58.0|
>
> And result is shown in table 5 on page 8 in the revised manuscript.
>
> Gaussian Mixture Model (GMM) has been proven as an efficient method to distinguish between "positive" (reliable) and "negative" (unreliable) samples from teacher module's prediction[1].
>
> [1].Xinjiang Wang, Xingyi Yang, Shilong Zhang, Yijiang Li, Litong Feng, Shijie Fang, Chengqi Lyu,
> Kai Chen, and Wayne Zhang. Consistent-teacher: Towards reducing inconsistent pseudo-targets
> in semi-supervised object detection. In Proceedings of the IEEE/CVF conference on computer
> vision and pattern recognition, pp. 3240–3249, 2023b
>
> > Weaknesses 3: Significant Contradiction in the Experimental Design and Claimed Contributions This is my most serious concern. The authors propose a new "Overall Sparse Method" for dataset creation as a contribution to overcome the bias of the existing "Single Sparse Method" (Lines 100-104, 375-377). However, the paper's own ablation study in Table 5 clearly shows that their proposed "Overall Sparse Method" is consistently and significantly inferior to the "Single Sparse Method" across all settings. More problematically, the authors then abandon their own proposed method and use the superior-performing "Single Sparse Method" for all main comparison experiments (Table 1, 2), citing "a fair comparison with prior studies" (Line 394). This creates a major logical flaw: a method introduced as a contribution is empirically proven to be worse and is subsequently discarded in the main experiments. A contribution that fails its own evaluation cannot be claimed as a strength of the paper and undermines the work's scientific rigor.
>
> We need to clarify the rationale for employing the "Single Sparse Method" in our comparative analysis.
>
> First，the inherent issue with the common Single Sparse Method (used by existing work): This method employ a heuristic where every category present in an image is mandatorily preserved by at least one instance. This design leads to a significant bias: categories that are naturally rare in the overall dataset (such as Baseball-diamond, Ground-track-field, Soccer-ball-field, and Roundabout in DOTA) are over-represented in the final sparse subset, as their few instances are disproportionately likely to be forcibly retained. Consequently, experiments based on a nominal 20% sparsity ratio do not genuinely reflect the model's performance on a true 20% sparse dataset.
>
> Second, to address this issue and accurately measure performance under a given sparsity ratio, we propose a more rigorous and unbiased sampling method based on Overall Sparse Method. Our method achieves the desired sparsity level by performing global sampling over the full set of ground truth annotations across all categories in the dataset, thereby providing a fairer and more accurate assessment of model performance at that specific level of data scarcity.
>
> Finally, to ensure that our performance gains are directly attributable to our proposed algorithmic improvements，we made a necessary compromise in our primary comparative analysis against state-of-the-art sparse learning models under "Single Sparse Method" setting.

---

### Official Review · Reviewer_gKiC · 2025-10-31

**Soundness:** 3
**Presentation:** 3
**Contribution:** 3
**Rating:** 6
**Confidence:** 5

**Summary:**

This paper presents a novel and promising setting (Sparse Partial Weakly-Supervised ) that makes significant strides in addressing the high annotation costs associated with oriented object detection in remote sensing imagery. Its innovations, methodology, and experimental results demonstrate considerable potential.

**Strengths:**

1. The paper introduces a novel Sparse Partial Weakly-supervised Oriented Object Detection (SPWOOD) framework designed to significantly reduce annotation costs for oriented object detection in remote sensing. This task setting is more challenging and can further reduce annotation costs.

2. SPWOOD effectively leverages sparsely annotated data, weakly annotated data, and a large volume of unlabeled data, supporting various annotation formats (RBox, HBox, Point) or their combinations. This flexibility is a significant advantage, allowing it to adapt to diverse annotation resources.

3. The author employs a “simplify and divide” approach to transform a seemingly intractable, contradictory learning task into a series of subproblems that can be tackled through technical means.

4. The paper is well-structured.

**Weaknesses:**

1. While attempting to classify different methods and their cost-effectiveness, it is overly dense.

2. The paper claims the Overall Sparse Method addresses the bias of the Single Sparse Method. However, Table 6 on page 9 shows that the Single Sparse Method sometimes yields higher annotation counts and AP performance for several categories, which appears contradictory to the paper's claims. While the paper attempts to explain this by stating the Single Sparse Method "tends to retain more annotations for categories with initially scarce labels," the explanation is not deep enough.

3. The description of the Multi-level Pseudo-label Filtering (MPF) implementation details is not sufficiently thorough. For example, specific details on the initialization and update mechanisms for GMM parameters (mean, variance, weights) are missing.

4. While the ablation study demonstrates MPF's superiority over CPF, a more detailed analysis of the individual contributions of different components within SPWOOD (e.g., orientation learning, scale learning, sparse annotation learning within SOS-Student) would be beneficial.

**Questions:**

1. I would like to know whether the symmetry assumption in directional learning limits the generalisation ability of the method.

2. The paper demonstrates SPWOOD's capability with a combination of annotation types (R:H:P=1:1:1 in Table 1). This is a realistic and strong setting. However, how does the framework's performance vary with different proportions of weak labels?

3. The ablation study on sparse data creation methods (Table 5 & 6) presents a very interesting, albeit counter-intuitive, finding: the "Single Sparse Method," which introduces sampling bias by over-representing rare categories, outperforms the more distributionally faithful "Overall Sparse Method." The paper attributes this to the raw annotation count. Could this also indicate that the model is simply overfitting to these rare classes, and its performance on common classes might be suffering? Furthermore, what are the practical implications of this finding?

4.  Is the performance gain of SPWOOD significant enough to justify its potential increase in computational cost, particularly for resource-constrained research environments?

---

> ### Author Response · Authors · 2025-11-21
>
> > Weaknesses 1: While attempting to classify different methods and their cost-effectiveness, it is overly dense.
>
> Thank you for pointing out that the current cost-effectiveness visualization (Figure 1) is overly dense. We acknowledge that the complexity makes it difficult to parse at a glance.
>
> This density is largely a result of the necessary trade-off between the various annotation formats and supervisory settings. Since our pipeline is designed to be highly complex and unified, reflecting this complexity in a single visualization is challenging, even though we have already simplified it significantly.
>
> However, we fully agree that clarity is paramount. As you kindly noted in the strengths section, our approach is capable of effectively “simplifying and dividing” complex concepts. Building upon this, we will dedicate further effort to refine the visualization.
>
> We commit to streamlining and simplifying Figure 1 in the revised manuscript to substantially enhance its clarity and readability.
>
> > Weaknesses 2: The paper claims the Overall Sparse Method addresses the bias of the Single Sparse Method. However, Table 6 on page 9 shows that the Single Sparse Method sometimes yields higher annotation counts and AP performance for several categories, which appears contradictory to the paper's claims. While the paper attempts to explain this by stating the Single Sparse Method "tends to retain more annotations for categories with initially scarce labels," the explanation is not deep enough.
>
> We must clarify that this is a misunderstanding.
>
> First，Single Sparse Method (used by existing work): This method employ a heuristic where every category present in an image is mandatorily preserved by at least one instance. This design leads to a significant bias: categories that are naturally rare in the overall dataset (such as Baseball-diamond, Ground-track-field, Soccer-ball-field, and Roundabout in DOTA) are over-represented in the final sparse subset, as their few instances are disproportionately likely to be forcibly retained.
>
> Second, Overall Sparse Method achieves the desired sparsity level by performing global sampling over the full set of ground truth annotations across all categories in the dataset, thereby providing a fairer and more accurate assessment of model performance at that specific level of data scarcity.
>
> As highlighted in Table 10 on page 10, Single Sparse Method obtains more annotations counts than Overall Sparse Method, especially for categories Baseball-diamond, Ground-track-field, Soccer-ball-field, and Roundabout, which result in extramly higher performance. Crucially, categories with similar annotation counts yield nearly identical results.

---

> ### Author Response · Authors · 2025-11-21
>
> > Weaknesses 3: The description of the Multi-level Pseudo-label Filtering (MPF) implementation details is not sufficiently thorough. For example, specific details on the initialization and update mechanisms for GMM parameters (mean, variance, weights) are missing.
>
> Thank you for your suggestion. In the revised manuscript, we will provide a substantially more detailed description of the MPF implementation.
>
> > Weaknesses 4: While the ablation study demonstrates MPF's superiority over CPF, a more detailed analysis of the individual contributions of different components within SPWOOD (e.g., orientation learning, scale learning, sparse annotation learning within SOS-Student) would be beneficial.
>
> We appreciate the reviewer's interest in a deeper component-wise analysis of the SOS-Student module within SPWOOD.
>
> We must first clarify a fundamental design aspect: the SOS-Student is intentionally designed as a unified and tightly coupled framework. Specifically, the orientation learning and scale learning modules are fundamentally inseparable from the core SOS-Student model. This is due to the design to robustly handle the weak annotations, like HBox and point annotations.
>
> Following your suggestion, we have conducted a dedicated ablation study focused on the impact of sparse annotation learning(SAL).
>
> | Setting | mAP |
> |----------|----|
> | with SAL | 47.6 |
> | without SAL|42.0|
>
> The results of this comprehensive analysis are also presented in Table 6 on page 8 of the revised manuscript.

---

> ### Author Response · Authors · 2025-11-21
>
> > Questions 1: I would like to know whether the symmetry assumption in directional learning limits the generalisation ability of the method.
>
> We must clarify that the symmetry assumption does not inherently limit the generalization ability of our method.
>
> While the assumption is useful, our models are not totally reliant on it. We acknowledge that many categories in complex datasets like DOTA are not strictly symmetrical (e.g., harbor, ship, or helicopter). In these cases, the model learns to adjust the initial symmetric prior.
>
> Therefore, the symmetry assumption acts primarily to enhance learning efficiency and stability in rotation-invariant scenarios, but does not impede the network's capacity to generalize to complex, asymmetric real-world objects.
>
> > Questions 2: The paper demonstrates SPWOOD's capability with a combination of annotation types (R:H:P=1:1:1 in Table 1). This is a realistic and strong setting. However, how does the framework's performance vary with different proportions of weak labels?
>
> Thank you for this suggestion. We conduct additional experiments about different conbinations ratio of weak labels. The RBox:HBox:Point ratio reflects the relatively proportion of each annotation type among all annotations.
>
> | RBox:HBox:Point| mAP |
> |----------|----|
> |     1:1:1| 53.0|
> |1:1:0| 56.3|
> |1:0:1| 52.3|
> |0:1:1| 47.6|
> |0:1:4| 41.2|
>
> The result is also shown in Table 4 on page 8 of the revised manuscript.

---

> ### Author Response · Authors · 2025-11-21
>
> > Questions 3: The ablation study on sparse data creation methods (Table 5 & 6) presents a very interesting, albeit counter-intuitive, finding: the "Single Sparse Method," which introduces sampling bias by over-representing rare categories, outperforms the more distributionally faithful "Overall Sparse Method." The paper attributes this to the raw annotation count. Could this also indicate that the model is simply overfitting to these rare classes, and its performance on common classes might be suffering? Furthermore, what are the practical implications of this finding?
>
> Thank you for this question.
>
> We must clarify that the performance gap is primarily due to underfitting rather than the model overfitting to rare classes. In the context of object detection, the number of training epochs is inherently limited, which prevents severe overfitting, especially given the data sparsity.
>
> - Overall Sparse Method: This method accurately reflects the scarcity, resulting in very low annotation counts for many classes. This severe data constraint, combined with limited training time, leads to underfitting, preventing the model from learning sufficient features, particularly for low-data categories.
> - Single Sparse Method: While biased, this method uses a heuristic to ensure a higher absolute number of preserved annotations across the dataset. This provides the model with enough necessary samples to move out of the underfitting regime, thus yielding a higher overall score.
>
> Regarding the concern that performance degradation in low-data categories might affect common classes:
>
> As highlighted in Table 10, our empirical evidence suggests that the performance degradation observed in low-data categories (when using the Overall Sparse Method) does not adversely impact the results achieved in high-data categories. The performance differences are localized to the categories most affected by the sampling strategy.
>
> The Overall Sparse Method provides a novel and essential approach to generating distributionally faithful sparsely-annotated datasets. Its practical implications are significant for future research:
> - It serves as a more rigorous and unbiased benchmark for evaluating the true efficacy of new sparse learning algorithms under specific, controlled scarcity levels.
> - It allows researchers to separate the performance gain derived from algorithmic innovation from the gain introduced by data sampling bias (heuristic design).
>
>
> > Questions 4: Is the performance gain of SPWOOD significant enough to justify its potential increase in computational cost, particularly for resource-constrained research environments?
>
> Thank you for this important question regarding the trade-off between performance gain and computational cost, especially in resource-constrained settings.
>
> We fully address this concern by providing a comprehensive comparison of our SPWOOD framework against related models, detailing metrics such as Training Time and GPU Memory consumption. The result is also presented in Table 11 on page 16 of the revised manuscript.
>
> | Algorithm Types | Methods | memory usage  | running time |
> |----------|----|----|----|
> |Semi-Supervised| MCL|     5598MB| 16hours|
> |Partial Weakly-Supervised| PWOOD|    9021MB| 23hours|
> |Sparsely Supervised| RSST|     13672MB| 34hours|
> |Sparse Partial Weakly-Supervised| SPWOOD|     22785MB| 40hours|
>
> We acknowledge that SPWOOD requires marginally higher computational resources and training time compared to baselines. However, we argue that this increase is justified and acceptable for the following reasons:
>
> As the first unified teacher-student framework that leverages sparse, weak, and unlabeled data, SPWOOD achieves state-of-the-art results. The marginal increase in computational complexity is a necessary investment to unlock superior performance.

---

### Author Response · Authors · 2025-11-26

Dear Reviewers,

We hope this message finds you well. We wanted to follow up on the rebuttal we submitted for our ICLR submission.

We know the review period keeps you busy, and we really appreciate the time you've put into reviewing our paper. We've worked hard to address the concerns from your review. Whenever you have a moment, we'd love to hear what you think about our responses.

If you need us to clarify anything else, just let us know. Thanks so much for your time.

Best regards,

Authors

---

### Author Response · Authors · 2025-12-03
**Thanks AC for taking over our submission and Summary of Review-Rebuttal phase**

Dear Area Chair,

Thank you for taking on this paper and for being invited to participate in our discussion with the reviewers. We understand that the special circumstances this year have placed an additional burden on ACs, and we genuinely appreciate your time and effort.

We are pleased to report that the core concerns have been resolved. Reviewer @JpkM (Rating 4, Confidence 3 -> Rating 6 after rebuttal) have confirmed that our response fully addresses queries and recommend acceptance. The revisions also provide substantial evidence to resolve the concerns of Reviewers @gKiC(Rating 6, Confidence 5), @2UVa(Rating 4, Confidence 4) and @Xk5r(Rating 8,Confidence 5).

Best regards,

The Authors

---

### Meta-Review · Area_Chair_3QcS · 2026-01-12

**Summary:**

Reviewers appreciated that the proposed framework is novel and effectively leverages sparsely annotated data, while the task setting is more challenging. Reviewers also appreciated that the manuscript is well-structured and the experimental results demonstrate the effectiveness of the proposed methodology. Reviewers also raised some concerns in the initial review including, missing discussion on similar sparse partial supervision paradigms, experiments beyond DOTA dataset to show generality, missing qualitative failure analysis, missing implementation details (sparse annotation learning, multi-level pseudo-labels filtering), and clarification regarding single sparse method

**Reviewer Concerns:**

Authors provided a comprehensive rebuttal to address the initial concerns raised by the reviewers. For instance, authors integrated a detailed discussion of related sparse supervision methods from other domains in the revised manuscript. To support the generality claim, authors provided additional experimental validation on the DIOR dataset. Authors also provided qualitative failure analysis section in Appendix A.3. Authors also provided details regarding the adaptive factor and clarifying the rationale for employing the single sparse method in their comparative analysis. Post-rebuttal, one of the reviewers (reviewer JpkM) mentioned to be satisfied with the author's response, raising the score to a positive rating.

**Reviewer Scores:**

The meta-reviewer believes most of the concerns of the the reviewers were addressed in the rebuttal. Three of the reviewers are generally on the positive side. For the remaining reviewer, authors provided response to all the concerns including, details regarding the adaptive factor and clarifying the rationale for employing the single sparse method in their comparative analysis. Given that most of the reviewers are on the positive side, a comprehensive rebuttal provided by the authors, the meta-reviewer believes the strength outweights weaknesses for this manuscript and agrees with the majority of the reviewer's positive ratings.

---

### Decision · Program_Chairs · 2026-01-26

Accept (Poster)